# Fabrication and Characterization of Novel Shape-Stabilized Phase Change Materials Based on P(TDA-*co*-HDA)/GO Composites

**DOI:** 10.3390/polym11071113

**Published:** 2019-07-01

**Authors:** Sai Chen, Yue Yu, Ruirui Cao, Haihui Liu, Xingxiang Zhang

**Affiliations:** Tianjin Municipal Key Lab of Advanced Fiber and Energy Storage Technology, School of Material Science and Engineering, Tianjin Polytechnic University, Tianjin 300387, China

**Keywords:** shape-stabilized PCMs, alkyl acrylate, thermal energy storage, graphene oxide

## Abstract

Shape-stabilized phase change materials (SPCMs) are green, reusable energy storage materials. Because the melting temperature of *n*-alkyl acrylate copolymer is adjustable by controlling the side-chain length, the appropriate melting temperature can be achieved. Poly(tetradecyl acrylate-*co*-hexadecyl acrylate) (P(TDA-*co*-HDA)) with a molar ratio of 1:1 and SPCMs were fabricated via an atom transfer radical polymerization (ATRP) method and a solution blending method with P(TDA-*co*-HDA) as a thermal storage material and graphene oxide (GO) as a supporting substance. In this composite, an SPCM was achieved, which absorbed heat at 29.9 °C and released it at 12.1 °C with a heat storage capacity of 70 J/g at a mass ratio of GO of 10%. The material retained its shape without any leakage at 60 °C, which was much higher than that of the melting temperature of P(TDA-*co*-HDA). The SPCMs exhibited good crystallization behaviors and excellent thermal reliabilities after 100 thermal cycles. The thermal properties of the P(TDA-*co*-HDA)/GO composite PCMs with various GO loadings were also investigated. The novel shape-stabilized PCMs fabricated in this study have potential uses in thermal energy storage applications.

## 1. Introduction

Increasing greenhouse gas emissions and fossil fuel consumption fueled the exploration of more efficient renewable energy sources [1,2]. Effective energy storage devices and systems are critical to increasing energy adequacy and reducing time and space mismatches in energy supply and demand, thereby reducing environmental impact [3,4,5,6]. Energy savings could be realized by thermal energy storage systems that use phase change materials (PCMs) [6,7,8]. PCMs are widely used in many fields, including vehicle battery energy management systems, energy saving buildings, thermal-regulated fibers and textiles, solar heat storage and waste heat recovery, heat transfer fluids, and thermal insulation materials [7,8,9,10,11,12]. However, there is an insufficient number of PCMs to meet the needs of growing applications.

Our group found that poly(*n*-alkyl acrylate)(PAA) with more than 10 atoms in its side-chain was a phase change material [13,14]. The melting temperature (*T*_m_) of the long side-chains, which melt during the melting process, can be controlled by the length of the side chain, causing significant changes in the physical properties of the polymer [15]. The temperature gap of PAA with an even adjacent carbon number in the side-chain is too high for building and textile applications. However, PAA with an odd number of carbons in the side chain is difficult to fabricate at the pilot scale. Furthermore, the direct utilization of poly(tetradecyl acrylate-*co*-hexadecyl acrylate)(P(TDA-*co*-HDA)) is restricted by the leakage of the liquid phase when used higher than the melting temperature. Therefore, shape-stabilized phase change materials (SPCMs) attracted extensive interest for latent heat energy storage in recent years.

Graphene received extensive attention in recent years due to its excellent properties such as high thermal conductivity, excellent mechanical properties, and excellent electronic properties [16,17,18,19,20]. Graphene-derived graphene oxide (GO) is a two-dimensional nanosheet skeleton material having oxygen-containing functional groups such as hydroxyl, carbonyl, and epoxy groups at its base and edges. The high porosity and surface area of graphene oxide allow it to interact with other organic materials, enabling the material to adhere to the liquid under the action of surface tension and capillary force, which is beneficial to the shape stability of PCMs during solid–liquid conversion [21,22].

In recent years, there were more and more researches on the preparation of hybrid composite PCMs with organic and inorganic materials as support matrix/shape stabilizers [23]. Inorganic porous materials, such as diatomite, silicon nanopowders, and carbon-based nanofillers, are widely used due to their strong physical adsorption capacity and large surface area [24,25,26,27].

In a previous study, Qi et al. [22] fabricated PEG/GO shape-stabilized PCMs by introducing GO as the supporting material using a blending method. Cao et al. [13] synthesized shape-stabilized composite PCMs via in situ free-radical polymerization. The composite included poly(hexadecyl acrylate) (PHDA)/GO and PHDA finding the synergistic phase change effect of PHDA-g-GO and PHDA.

In this study, we firstly prepared the P(TDA-*co*-HDA) (molar ratio 1:1) via atom transfer free radical polymerization (ATRP). We subsequently fabricated P(TDA-*co*-HDA)-based shape-stabilized PCMs by introducing GO as the supporting material using a simple solution blending method. This fabricated novel shape-stabilized PCMs (P(TDA-*co*-HDA)/GO nanocomposites) are named SPCMs. The morphological variations, phase change behaviors, thermal reliabilities and stabilities, and crystalline properties of the composite materials were investigated.

## 2. Experimental

### 2.1. Materials

Tetradecyl acrylate (TDA) and hexadecyl acrylate (HDA) were bought from TCI (Shanghai, China) Development Co., Ltd. Ethyl 2-bromopropionate (98%, EBP), triphenylphosphine (>99%, PPh_3_), and iron chloride tetrahydrate (99.95%, FeCl_2_·4H_2_O) were provided by Aladdin Reagent (Shanghai, China). Natural graphite powders (325 mesh) were offered by Qingdao Laixi Graphite Co. Ltd. (Qingdao, China) and used as received. Sulfuric acid (H_2_SO_4_, 98%), potassium permanganate (KMnO_4_, 99.3%), hydrogen peroxide (H_2_O_2_, 30%), hydrochloric acid (HCl, 37%), aluminum oxide(Al_2_O_3_, AR), *N,N*-dimethylformamide (DMF, AR), toluene (AR), tetrahydrofuran(THF, AR), and methanol (AR) were afforded by Guangfu Fine Chemical Research Institute (Tianjin, China). The PPh_3_ was recrystallized and the toluene was refluxed with sodium metal when it was used.

### 2.2. Synthesis of P(TDA-co-HDA)

P(TDA-*co*-HDA) was fabricated by ATRP. The general polymerization procedure was as follows: TDA (13.42 g, 0.05 mol)), HDA (14.83 g, 0.05 mol), toluene (21.20 mL), PPh_3_ (0.79 g, 3 × 10^−3^ mol), and FeCl_2_·4H_2_O (0.30 g, 1.5 × 10^−3^ mol) were firstly added to a 250-mL Schlenk flask in turn, after which the flask was sealed. After freezing, the flask was evacuated and its contents were melted twice, and EBP (128 uL, 1 × 10^−3^ mol) was poured into the Schlenk flask with a 1-mL injection syringe, which was repeated. The reaction system was then placed into an oil bath at 100 °C under magnetic stirring for 12 h. After cooling to room temperature, the resultant solution was purified by aluminum oxide. The obtained suspension was precipitated in a large amount of methanol. The purified precipitate was subsequently dried in a vacuum oven at 35 °C for 24 h. The final product was P(TDA-*co*-HDA). Poly(tetradecyl acrylate) (PTDA) and PHDA were obtained in the same manner for comparison.

### 2.3. Fabrication of SPCMs

GO powders were synthesized by the modified Hummers method, which was reported in our previous work [28,29]. In this experiment, concentrated H_2_SO_4_ (200 mL) was added to a mixture of natural graphite (8 g) and NaNO_3_ (20 g). The mixture was stirred in ice water. KMnO_4_ (36 g) was slowly added, and stirred to maintain the reaction temperature below 10 °C. Then, the reaction was carried out at 35 °C, stirring for 5 h. Afterward, deionized water (500 mL, 70 °C) was added dropwise. The temperature of the reaction system was then cooled to room temperature, and 30% H_2_O_2_ (10 mL) was added. The color of the mixture turned bright yellow when the reaction was over. The GO was dispersed in a 4% HCl solution and washed five times repeatedly. The product was then further washed with deionized water to completely remove the metal ions and acid until the pH was neutral. Finally, the GO solution was freeze-dried to obtain a brown fluffy GO product.

P(TDA-*co*-HDA)/GO composites were fabricated via a simple physical solution blending method. GO powders were firstly dispersed in DMF with ultrasonication for 1 h to form a homogeneous suspension. The GO suspension was subsequently dripped into a P(TDA-*co*-HDA) toluene solution with vigorous stirring at 75 °C for 12 h. Finally, the product was put in a vacuum oven at 35 °C. The loadings of GO in the P(TDA-*co*-HDA)/GO composites were 0, 2, 4, 6, 8, 10, and 15 wt.%, and the obtained products were named P(TDA-*co*-HDA), SPCM2, SPCM4, SPCM6, SPCM8, SPCM10, and SPCM15, respectively.

Scheme 1 shows the synthesis illustration for the P(TDA-*co*-HDA) and SPCMs via the ATRP and solution blending methods.

### 2.4. Characterization

Fourier-transform infrared (FTIR) spectra of the specimens were studied through a spectrometer (BrukerTERSOR37, Karlsruhe, Germany) in the range of 4000 to 400 cm^−1^. The specimens were recorded on a KBr disk at 4-cm^−1^ resolution.

Micro-Raman mapping spectra were recorded on a Raman microscope (XPLORA PLUS, Kyoto, Japan) equipped with a 532-nm laser source.

The number-average molecular weight (*M*_n_) and weight-average molecular weight (*M*_w_) of P(TDA-*co*-HDA) were obtained by gel permeation chromatography (GPC, Viscotek 270, Malvin, USA) in THF at room temperature.

The surface morphologies of the SPCMs were observed by field-emission scanning electron microscopy (FE-SEM, Hitachi S-4800, Tokyo, Japan).

The microstructures of the SPCMs were performed using transmission electron microscopy (TEM, Hitachi H-7650, Tokyo, Japan).

X-ray diffraction (XRD) patterns of the SPCMs were characterized by a diffractometer (Rigaku D/MAX-gA, Tokyo, Japan) with filtered Cu *K*α radiation(λ = 0.15406 nm). Samples were scanned in the range from 3° to 40° (2θ), with a scan speed of 8°/min at room temperature.

Differential scanning calorimetry (DSC, NETZSCH 200 F3, Bavaria, Germany) was applied for phase change properties studies. Firstly, 5 to 10 mg of a specimen was encapsulated in an aluminum pan under a nitrogen atmosphere and heated from −20 to 60 °C at a rate of 10 °C/min, then kept at 60 °C for 2 min. Subsequently, the specimen was cooled to −20 °C at a rate of −10 °C/min and maintained for 2 min. Finally, the specimen was heated again from −20 to 60 °C at a rate of 10 °C/min. DSC thermograms in the first cooling and second heating processes were recorded.

Shape stabilities of the SPCMs were tested by visual observations through an oven and a digital camera. The SPCMs were put in an oven at 60 °C, which is high above the melting temperature of P(TDA-*co*-HDA), for 30 min.

Thermal stabilities of P(TDA-*co*-HDA) and SPCMs were obtained using thermogravimetric analysis (TG, NETZSCH STA409PC, Bavaria, Germany) from 30 °C to 600 °C with a heating rate of 10 °C/min under a nitrogen atmosphere.

## 3. Results and Discussion

### 3.1. Morphologies and Chemical Structures of SPCMs

The *M*_n_ and polydispersity index values of P(TDA-*co*-HDA)were measured to be 13,713 g/mol and 1.45, respectively.

FTIR spectra of the P(TDA-*co*-HDA) and SPCMs are shown in Figure 1.Characteristic bands of GO appeared at 3434 cm^−1^ (C–OH stretching), 1737 cm^−1^ (C=O stretching), 1634 cm^−1^ (C=C stretching vibration), and 1057 cm^−1^ (C–O of epoxy stretching). For P(TDA-*co*-HDA), characteristic bands were present at 1737 cm^−1^ (C=O stretching) and 1250 and 1176 cm^−1^ (C–C stretching of the alkyl chain in P(TDA-*co*-HDA)). In the spectra of the SPCMs, the majority of the absorption peaks of the primary functional groups of P(TDA-*co*-HDA) and GO also appeared, which had a slight shift in the peak positions, which proved that the SPCMs were successfully obtained.

The structural disorder and relative intensity (ID/IG) ratio of D and G bands were studied by micro-Raman spectroscopy. The D band appears due to the vibrations of *sp*^3^-bonded carbon atoms (defects), and the G band arises from the vibrations of *sp*^2^-bonded carbon atoms of the graphene rings [13,14]. SPCM10 was taken as an example. Figure 2 shows the micro-Raman spectra of GO and SPCM10. The samples displayed a strong D band at ~1360 cm^−1^ and a strong G band at ~1580 cm^−1^. The ID/IG ratios of GO and PCM10 were 0.930 and 0.937, respectively. Therefore, we can confirm that, after blending the P(TDA-*co*-HDA), the GO structure showed nearly no change.

The TEM micrographs of GO and the SPCM6 are shown in Figure 3. The transparent and lamellar morphology of GO can be observed. Figure 3b,c show that the transparency decreased upon the addition of P(TDA-*co*-HDA) owing to the absorbance of PCM on the surface of GO.

Figure 4 shows the SEM micrographs of (a) GO, (b) SPCM8, (c) SPCM10, and (d) SPCM15. The GO sheet exhibited wrinkled surface textures with curled edges. This structure played an important role in strengthening the interlocking of sheets and enabled strong interactions with the P(TDA-*co*-HDA). The GO sheets were homogeneously dispersed in the SPCMs. With the increased loading of GO, SPCMs exhibited a layered packing structure little by little, which was similar to GO due to the strong adsorption and interaction of P(TDA-*co*-HDA) with the GO sheets.

### 3.2. Thermal and Crystalline Properties of SPCMs

Figure 5 shows the DSC heating and cooling curves of pristine PTDA, PHDA, P(TDA-*co*-HDA), and SPCMs with various loadings of GO. The appropriate melting temperature of *n*-alkyl acrylate copolymer could be achieved by controlling the side-chain length. The fabricated SPCMs exhibited obvious endothermic/exothermic peaks. The detailed parameters are listed in Table 1. Solid–liquid phase transition of P(TDA-*co*-HDA) absorbed 99 J/g of heat at 21.2 °C with a peak temperature at 30.8 °C. During the crystallization process, the phase change process released 98 J/g of heat at 20.4 °C with a peak temperature of 13.4 °C. Compared with P(TDA-*co*-HDA), the *T*_mo_, *T*_co_, and *T*_mp_ values of the SPCMs did not change significantly, while the *T*_cp_ rose with the loading of GO and subsequently went down. It was reported that the doped fillers act as the heterogeneous nucleation seeds that result in a significant increase in *T*_c_ in micro-filled nanocomposites [30,31]. Nevertheless, at a relatively high loading of GO, although there were more heterogeneous nucleation sites that appeared, the high specific surface area of the two-dimensional GO sheets could limit the movement of the P(TDA-*co*-HDA) chains during crystallization because of the strong hydrogen-bond interactions between GO and P(TDA-*co*-HDA) [22]. Therefore, the high amount of GO resulted in a decrease in *T*_c_.

Table 1 shows that the Δ*H*_m_ and Δ*H*_c_ values of the SPCMs decreased with increased loading of GO, which was caused by the addition of GO, which did not undergo a phase change. The higher the loadings of GO were in the composite, the lower the phase transition enthalpy was. In addition, due to the restrictions of strong intermolecular hydrogen bonds, the GO sheets hindered the regular arrangement of the P(TDA-*co*-HDA) chains into the crystal lattice, resulting in a decrease in the phase transition enthalpy. The *T*_mp_ values of the SPCMs were close to the temperature of human skin, which may allow the application of the SPCMs in smart textiles.

The XRD patterns of GO, P(TDA-*co*-HDA), and the SPCMs are shown in Figure 6. The typical diffraction peak for the GO sheets appeared at approximately 11.21 (2θ), corresponding to the (001) reflection. The derived *d*-spacing was about 0.789 nm based on the Bragg equation, which was in good agreement with the value reported in the literature [2]. The P(TDA-*co*-HDA) exhibited a typical (110) diffraction peak at 21.28 (2θ), which implied that P(TDA-*co*-HDA)possessed good crystallization properties. The XRD patterns of the SPCMs showed the diffraction peaks of P(TDA-*co*-HDA) and GO after the GO was dispersed into the PCM matrix, implying that the crystallization behaviors of the SPCMs were well maintained and the GO sheets were homogeneously distributed in the P(TDA-*co*-HDA) matrix. As shown in Figure 5, an observable difference was that the SPCMs exhibited the (001) diffraction peak of GO at about 9.32–9.90° (2θ).The calculated *d*-spacing of the GO was 0.893–0.948 nm, which was bigger than that of GO. The increase in the distance between the interlayers of GO nanosheets was attributed to the physical adsorption of the GO layers.

### 3.3. Shape-Stabilized Properties of SPCMs

The shape-stabilized properties are important parameters of thermal energy storage materials on account of them preventing PCM leakage and broadening the application areas. Figure 7 shows the different morphologies of the SPCMs. The SPCMs exhibited a leakage phenomenon when the GO loading was below 10 wt.%. Until the GO loading reached 10 wt.%, the SPCMs maintained their shape, which indicates that the total amount of added GO was relatively low [32]. However, the loading was much higher than that of poly(hexadecyl acrylate-*co*-GO) [13], in which it was as low as 2 wt.%. The effect of covalent bonds on preventing molecular chain formation was much stronger than that of van der Waals forces. The strong adsorption of GO and strong intermolecular hydrogen bonds between P(TDA-*co*-HDA)and GO sheets contributed to the shape stabilization of PCMs. The shape-stabilized properties of the SPCMs could overcome the drawback of solid-to-liquid PCM leakage. Hence, the SPCMs have good practicability under the appropriate amount of GO.

### 3.4. Thermal Reliabilities and Structural Stabilities of the SPCMs

The thermal reliability and cycling stability of a PCM are considered to be important parameters for thermal energy storage applications. For SPCM10, as an example, after one and 100 thermal cycles, the thermal reliabilities and structural stabilities were compared. The DSC curves and FTIR spectra of SPCM10 are shown in Figure 8a,b, respectively. The detailed calorimetric data are summarized in Table 2. No changes were found in Figure 8a after one and 100 thermal cycles, which proved that SPCM10 possessed good thermal reliability. The *T*_m_, *T*_c_, Δ*H*_m_, and Δ*H*_c_ values of SPCM10 were nearly the same before and after 100 thermal cycles, as shown in Table 2. Although the data exhibited a slight change, these changes were negligible for thermal energy storage applications. The results demonstrated that the fabricated SPCMs also possessed good thermal reliabilities. Moreover, no structural changes were detected after the thermal cycling, as shown in Figure 8b. This indicated that the SPCMs possessed good structural stabilities, which would allow them to be used as shape-stabilized thermal energy storage materials.

### 3.5. Thermal Stabilities of SPCMs

Figure 9 shows the thermal stabilities of GO, P(TDA-*co*-HDA), and the SPCMs. At 200–300 °C and 300–550 °C, two mass loss stages were observed. The first stage below 300 °C was due to the decompositions of the labile groups of GO, and the second stage above 300 °C was due to the major decompositions of the high-molecular-weight P(TDA-*co*-HDA). Although the thermal stabilities of the prepared SPCMs were lower than that of P(TDA-*co*-HDA) under the same experimental conditions, the prepared SPCMs had better thermal stabilities at the temperature nearly 200 °C higher than their working temperature. Therefore, the fabricated SPCMs possessed good thermal stabilities and the SPCMs have great application potential in thermal energy storage systems.

## 4. Conclusions

The thermal properties of P(TDA-*co*-HDA)/GO composite PCMs with various GO loadings were investigated. The melting temperature of acrylate copolymerreached an appropriate temperature by controlling the side-chain length. P(TDA-*co*-HDA) with a molar ratio of 1:1, and SPCMs were successfully prepared via an atom transfer radical polymerization method and a solution blending method with P(TDA-*co*-HDA) as a thermal storage material and GO as a supporting substance. The melting and freezing points of this composite were 29.9 and 12.1 °C, respectively, with a heat storage capacity of 70 J/g when the mass ratio of GO was 10%. This material retained its shape without any leakage when the temperature was above the melting point of P(TDA-*co*-HDA). Moreover, the SPCMs exhibited obvious crystallization behaviors and excellent thermal reliabilitiesafter 100 thermal cycles. The thermal properties of the P(TDA-*co*-HDA)/GO composite PCMs with various GO loadings were also investigated. The fabricated composite PCM is a very promising shape-stabilized PCM for applications in thermal energy storage.

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
