# Peer review of "Fabrication and Characterization of Novel Shape-Stabilized Phase Change Materials Based on P(TDA-co-HDA)/GO Composites"

_polymers, 2019, doi:10.3390/polym11071113_

Round 1
Reviewer 1 Report
The paper reports preparation and characterization of shape-stabilized phase change materials based on P(TDA-co-HDA)/GO materials. Such shape-stabilized phase change materials (SPCMs) are important and modern materials due to their potential and practical applications as reusable energy storage materials. Results of studies on such composites are published in peer reviewed prestigious journals. Therefore, the topic as well as its significance falls in the scope of Polymers. The paper presents valuable results but especially its presentation must be improved. The Introduction is short but based on 32 up-to-date papers it presents the most important tasks concerning phase change materials and acrylate based polymers. I would like to ask the Authors to complete/explain their statement from line 40 (“PAA with an odd number of carbons in the side chain is difficult to fabricate at the pilot scale.” Why?) The most important question concern lines 176-182 (see below).
Experimental:
Line 84: I think 128 L is wrong. I presume that should be “128 mL”.
Line 88: What is the time of drying?
Scheme 1: must be improved – please take care about clarity of the picture and proper positions of labels.
Line 117: “studyed” should be “studied”. Please supply also the technique used for data collection: ATR, KBr, …
Line 130: “…was studied to the phase change properties” consider “…was applied for phase change properties studies”
Results and discussion
Line 147: You observe very broad peak at 3434 cm-1. It is typical for water. Is it possible in your case. From my experience OH bands are usually much sharper.
Line 176-182: “the Tcp rised with the loading of GO and subsequently went down.” and further discussion. I cannot see this trend: 13.4, 12.0, 13.3, 17.0, 12.7, 9.8, 17.0. This trend is very often broken. the observed changes are irregular. The temperature change also in rather narrow range being 7.2°C (however, this range is a bit broader than for Tmo, Tmp, Tco: 4.6, 5.6, 2.5°C, respectively). You used SPCM10 for further studies but did you consider why the Tcp is significantly different and reaches such low temperature? For SPCM8 and SPCM15 these values are much higher.
Line 207: “9.32–99.90° (2θ)” the latter value is very improbable. Should it be 9.90?
Line 219-220: “The effect of covalent bonds on the prevention of molecular chain formation was much 219 stronger than that of van der Waals forces.” it is unclear. What covalent bonds do you think about, formed between acrylate and GO? You mentioned about physical absorption of GO (line 209). Are you referring to literature data [33]? Please rewrite this sentence.
Line 262-264: “The first stage below 300°C was due to the decompositions of the labile groups of GO, and the second stage above 300°C was due to the major decompositions of the high molecular weight P(TDA-co-HDA).” How can you explain that the highest thermal stability of SPCM10 and SPCM15 for the highest GO load. They show the highest decomposition temperature with process running practically in one step occurs? I would expect that for this sample two steps should be clearly visible due to 1. The significant load of GO and 2. its decomposition observed at lower temperature.
Author Response
Point 1: I would like to ask the Authors to complete/explain their statement from line 40 (“PAA with an odd number of carbons in the side chain is difficult to fabricate at the pilot scale.” Why?)
Response 1: When the side chain of PAA is odd number, the experimental materials are expensive and the experimental process is complicated. The side chain with even number of PAA is usually applied in laboratory experiment.
Point 2: Line 84: I think 128 L is wrong. I presume that should be “128 mL”.
Response 2: I have corrected it in my paper that “128 L” should be “128 uL”.
Point 3: Line 88: What is the time of drying?
Response 3: The purified precipitate was subsequently dried in a vacuum oven at 35°C for 24 h.
Point 4: Scheme 1: must be improved – please take care about clarity of the picture and proper positions of labels.
Response 4: I have corrected it in my paper that the scheme 1 is the following picture
Scheme 1. Schematic representation of the synthesis of the P(TDA-co-HDA) and SPCMs via the atom transfer radical polymerization and solution blending methods
Point 5: “studyed” should be “studied”. Please supply also the technique used for data collection: ATR, KBr.
Response 5: I have corrected it in my paper that “studyed” should be “studied”. The specimens were recorded on a KBr disk at 4 cm-1 resolution.
Point 6: “…was studied to the phase change properties” consider “…was applied for phase change properties studies”
Response 6: I have corrected it in my paper. Thank you for your advice.
Point 7: You observe very broad peak at 3434 cm-1. It is typical for water. Is it possible in your case. From my experience OH bands are usually much sharper.
Response 7: The GO powers have thorough dried that before used. At 3434 cm-1, the peak is a little broad possibly owing to the addition of GO is a little too much when test. This phenomenon also appeared in other literatures.
Point 8: “the Tcp rised with the loading of GO and subsequently went down.” and further discussion. I cannot see this trend: 13.4, 12.0, 13.3, 17.0, 12.7, 9.8, 17.0. This trend is very often broken the observed changes are irregular. The temperature change also in rather narrow range being 7.2 °C (however, this range is a bit broader than for Tmo, Tmp, Tco: 4.6, 5.6, 2.5°C, respectively). You used SPCM10 for further studies but did you consider why the Tcp is significantly different and reaches such low temperature? For SPCM8 and SPCM15 these values are much higher.
Response 8: Thank you very much for asking this question. I have retest the DSC of the SPCMs. The specific figure and data show in the blew.
Figure 1.DSC curves of PTDA, PHDA, P(TDA-co-HDA), and the SPCMs during the cooling processes.
Table 1.Phase change properties of PTDA, PHDA, P(TDA-co-HDA), and the SPCMs from the DSC analysis*.
Samples | ΔHm/Jg-1 | Tmo/°C | Tmp/°C | ΔHc/Jg-1 | Tco/°C | Tcp/°C |
PTDA | 72 | 14.8 | 22.3 | -71 | 12.3 | 6.2 |
PHDA | 109 | 21.7 | 36.4 | -109 | 34.1 | 24.6 |
P(TDA-co-HDA) | 99 | 21.2 | 30.8 | -98 | 20.4 | 13.4 |
SPCM2 | 96 | 21.7 | 29.6 | -96 | 20.5 | 13.5 |
SPCM4 | 90 | 23.4 | 30.8 | -90 | 20.6 | 13.9 |
SPCM6 | 81 | 22.1 | 35.2 | -80 | 22.8 | 15.8 |
SPCM8 | 75 | 23.0 | 34.1 | -75 | 22.1 | 13.1 |
SPCM10 | 70 | 20.0 | 29.9 | -70 | 20.5 | 12.1 |
SPCM15 | 67 | 24.6 | 32.0 | -67 | 22.9 | 13.3 |
*Tmo-onset melting temperature on the DSC heating curve; Tmp-peak melting temperature on the DSC heating curve; Tco-onset crystallizing temperature on the crystallization curve; Tcp-peak crystallizing temperature on the crystallization curve.
Point 9: “9.32–99.90° (2θ)” the latter value is very improbable. Should it be 9.90?
Response 9: I have corrected it in my paper that “99.90°” should be “9.90°”
Point 10: “The effect of covalent bonds on the prevention of molecular chain formation was much stronger than that of van der Waals forces.” it is unclear. What covalent bonds do you think about, formed between acrylate and GO? You mentioned about physical absorption of GO (line 209). Are you referring to literature data [33]? Please rewrite this sentence.
Response 10: In our previous work, PHDA-GO formed covalent bonds contributed to the shape-stabilization of PCMs, which needed a little addition of GO. P(TDA-co-HDA)/GO formed the van der Waals forces that required a little more GO. The strong adsorption of GO and strong intermolecular hydrogen bonds between P(TDA-co-HDA) and GO sheets contribute to the shape-stabilization of PCMs.
Point 11: “The first stage below 300°C was due to the decompositions of the labile groups of GO, and the second stage above 300°C was due to the major decompositions of the high molecular weight P(TDA-co-HDA).” How can you explain that the highest thermal stability of SPCM10 and SPCM15 for the highest GO load. They show the highest decomposition temperature with process running practically in one step occurs? I would expect that for this sample two steps should be clearly visible due to 1. The significant load of GO and 2. its decomposition observed at lower temperature.
Response 11: I think when the addition of GO increase, the first stage is higher. When the addition of GO achieve a critical value, the thermal stabilities of the SPCMs do not change too much. The SPCMs have better thermal stabilities at the temperature nearly 200°C higher than their working temperature.

Reviewer 2 Report
In this study by Chen et al. the authors have developed a phase change material with graphene oxide as a filler. The presented work is interesting and carried out well in general, but there are some shortcomings, which have to be addressed before recommending it for publication. Please find my suggestions below:
1) There are some minor spelling mistakes such as "studyed" (Line 117). Please proofread the manuscript and make appropriate corrections.
2) It would be good to include at least a short paragraph on the preparation of graphene oxide. There is a reference to your other works, but some information should be provided here as well.
3) It is not sufficient to simply state that graphene oxide is used without providing any characterization of this material. Please include at least Raman spectra to enable readers to judge the level of crystalinity, etc.
4) Scheme 1 is very much mixed up and have to be corrected. It is distorted, some parts overlap and some of carbon atoms in your graphene oxide schematics have five bonds, which is unacceptable, I am afraid.
Author Response
Point 1: There are some minor spelling mistakes such as "studyed" (Line 117). Please proofread the manuscript and make appropriate corrections.
Response 1: I have corrected it in my paper that “studyed” should be “studied”.
Point 2: It would be good to include at least a short paragraph on the preparation of graphene oxide. There is a reference to your other works, but some information should be provided here as well.
Response 2: I have added a short paragraph on the preparation of GO in the paper as follows.
In this experiment, concentrated H2SO4 (200 mL) was added to a mixture of natural graphite (8 g) and NaNO3 (20 g). The mixture was stirred in ice water. KMnO4 (36 g) was slowly added, stirred to maintain the reaction temperature below 10 °C. And then the reaction was carried out at 35 °C, stirring for 5 h. Afterwards, deionized water (500 mL, 70 °C) was added dropwise. The temperature of the reaction system was then cooled to room temperature, and 30% H2O2 (10 mL) was added. The color of the mixture turned bright yellow and the reaction was over. The GO was dispersed in a 4% HCl solution and washed 5 times repeatedly. The product was then further washed with deionized water to completely remove the metal ions and acid until the pH was neutral. Finally, the GO solution was freeze-dried to obtain a brown fluffy GO product.
Point 3: It is not sufficient to simply state that graphene oxide is used without providing any characterization of this material. Please include at least Raman spectra to enable readers to judge the level of crystalinity, etc.
Response 3: I have supply the Raman in the paper as follows.
Figure 1. Raman spectra of GO and the SPCM10.
The structural disorder and relative intensity (ID/IG) ratio of D and G bands were studied by micro-Raman spectroscopy. The D band appears due to the vibrations of sp3-bonded carbon atoms (defects), and the G band arises from the vibrations of sp2-bonded carbon atoms of the graphene rings[13-14]. SPCM10 takes an example. Fig. 2 shows the m-Raman spectra of GO and SPCM10. The samples display a strong D band at ~1360 cm-1 and a strong G band at ~1580 cm-1. The ID/IG ratios of GO and PCM10 are 0.930 and 0.937, respectively. Therefore, we can confirm that after blending the P(TDA-co-HDA), the GO structure nearly no change.
Point 4: Scheme 1 is very much mixed up and have to be corrected. It is distorted, some parts overlap and some of carbon atoms in your graphene oxide schematics have five bonds, which is unacceptable, I am afraid.
Response 4: I have corrected it in my paper that the scheme 1 is the following picture.
Scheme 1. Schematic representation of the synthesis of the P(TDA-co-HDA) and SPCMs via the atom transfer radical polymerization and solution blending methods.

Round 2
Reviewer 1 Report
The Authors referred to all doubtful or questionable points. Their answers are convincing and now I can recommend to accept this paper in the present form. This paper falls very well into the scope of this Journal.
Reviewer 2 Report
Thank you for including most of my suggestions (Scheme 1 is still distorted). I can now recommend publication of the article.